# Research on urban resilience assessment and driving forces based on Production-Living-Ecological Spaces: A case study of the Three Gorges Reservoir Area in China

**Honglin Liu**[1,2]*, **Junjie Zhu**[1,2], **Ziming He**[1,2]

**1** College of Civil Engineering & Architecture, China Three Gorges University, Yichang, China, **2** Hubei Engineering Technology Research Center for Disaster Prevention and Mitigation, China Three Gorges University, Yichang, China

* liuhonglin@ctgu.edu.cn

## Abstract

This study aims to develop a resilience evaluation approach based on production-living-ecological spaces and identify the key drivers of urban resilience in less-developed areas, taking the Three Gorges Reservoir Area in China as a case study. Understanding how to increase urban resilience and how to coordinate the three types of space in urban areas is critical for the healthy and stable development of cities. In this work, we develop an integrated assessment methodology for resilience in three dimensions: production space, living space, and ecological space. The geographical agglomeration impact is studied using the spatial approach, such as the Jenks natural breakpoint method and spatial autocorrelation analysis. Geographic Detectors are used to identify the drivers and to provide tailored optimization solutions. It was discovered that: (1) from 2010 to 2020, the comprehensive resilience of counties in the Three Gorges Reservoir Area of China continues to increase, as does the sub-spatial resilience; (2) both the resilience of counties and the sub-spatial resilience exhibit "unbalanced" characteristics, and the spatial distribution exhibits "randomization". There appears to be no evident clustering influence; (3) each county's production space has a high level of resilience, which is the component that has the greatest influence on the county's total resilience and the living space is next, followed by the ecological space; (4) the vast majority of interactions between the driving factors are two-factor enhancement, with just a small percentage being non-linear enhancement. The study's findings can provide theoretical guidance and technical help for the study of urban resilience in less-developed locations. Furthermore, the findings are extremely important for the development of resilient cities.

**Data availability statement:** All data files are available from a public repository and are open access. Given the wide variety of data used in this study, the URLs of the data sources have been separately stored in the Supporting Information Files.

**Funding:** Initials of the author who received award: HL Grant numbers:52078193 Funder name: the National Natural Science Foundation of China URL of funder website: https://www.nsfc.gov.cn/ The funder plays an important role in the study design, data collection and analysis, decision to publish, and preparation of the manuscript.

**Competing interests:** The authors have declared that no competing interests exist.

## 1. Introduction

Rapid urbanization has resulted in a rise in population, a decline in urban resources, a high concentration of urban activities, and a variety of natural catastrophes and public safety issues, all of which have adversely impacted city stability and the healthy lives of their residents [1]. For example, the COVID-19 outbreak in 2019 swept over the entire globe, posing a significant threat to global economic development [2]. Heavy rains in Zhengzhou, Henan, China, in 2021 killed hundreds of people and caused extensive property damage [3]. With China's urbanization rate reaching 64.72%, new difficulties, such as increased risk of natural disasters and fragile urban environments have emerged, requiring attention to the new hotspot of developing resilient cities [4,5]. The Proposals of the Central Committee of the Communist Party of China on Formulating the 14th Five-Year Plan for National Economic and Social Development and the Long-Range Objectives through the Year 2035, the United Nations 2030 Sustainable Development Goals (SDGs), and the New Urban Agenda all explicitly address the issue of sustainable urban development, emphasizing the construction of resilient cities to enhance urban sustainability [6]. The term "resilience" was taken from the Latin word "resilio", which originally meant "to return to the original state", and has been used since. The notion has been applied to a variety of sectors, including psychology, mechanical engineering, ecology, catastrophe prevention and mitigation, and so on, progressing from "engineering resilience" to "ecological resilience" and finally to "evolutionary resilience" [7–9]. The core concept is to improve cities' ability to withstand disruptions, adapt to unpredictable hazards, and swiftly return to normalcy, which is referred to as "urban resilience" [10]. As a result, it is critical to construct "resilient cities" in order to improve the ability to deal with risk concerns.

Terms like "studying urban resilience" and "building resilient cities" have been frequent in academic conversations about urban development in recent years, and certain outcomes have been obtained. The more developed coastal areas are the main focus of the study area for the assessment of urban resilience, but the natural conditions, spatial constraints, risk factors, and other things between cities in less developed mountainous areas and cities in developed coastal areas are very different, so the resilience evaluation methods of cities in developed coastal areas are not applicable to the resilience evaluation of cities in less developed mountainous areas [11]. Scholars have researched resilient cities from the standpoint of particular catastrophe response [12], climate change [13], or a single dimension of economic resilience [14], social resilience [15], ecological resilience [16], and infrastructure [17]. To examine resilient cities, some researchers employ disaster indicator system [18], system dynamics model [19], scale-density-morphology assessment model [20], and multi-scenario model simulation [21]. The dominant research approaches for evaluating resilient cities include the entropy weight method [22], hierarchical analysis [23], principal component analysis [24], and function method [25]. In addition, Huang et al. found that cities with a higher proportion of the tertiary sector and more diversified economic foundations have higher levels of economic resilience [14]. Norris et al. found that communities with strong social networks and a sense of collective efficacy

are more capable of withstanding disasters [26]. Moradi et al. found that cities with higher levels of green infrastructure (such as parks and green spaces) have better environmental resilience [27]. Khan et al. found that cities with advanced infrastructure systems have higher levels of resilience [28]. Casali, Y. et al. revealed the evolutionary patterns of cities over different time periods and their impact on urban resilience through the analysis of three subsystems: infrastructure, socio-economic conditions, and the built environment, over multiple years [29].

In general, urban resilience research is fast evolving and growing. However, there are still gaps in the following areas: In terms of research areas, China, as a developing country, still has many cities that are developing slowly and require in-depth exploration of urban resilience, and the Three Gorges Reservoir Area is a typical mountainous underdeveloped area that is an important ecological barrier area in the Yangtze River's middle and upper reaches, as well as an important economic growth point in China's Yangtze River economic belt. As a result, it is critical to design an urban resilience evaluation approach that is suitable for mountainous undeveloped locations. From the perspective of research, studies on developed areas from a single viewpoint have become quite mature. However, research at the regional level in less developed areas, which involves multiple dimensions, is relatively lacking. Few academics have utilized the complete evaluation approach to build a spatial evaluation system and then calculate the urban resilience index in terms of evaluation models and procedures. The comprehensive evaluation approach is an expansion of the entropy weight method that takes into consideration both indicator variability and the computation of comprehensive resilience.

Based on existing research on urban resilience, this paper proposes the concept of production-living-ecological spatial resilience: improving the absorption and digestion capacity of cities through the three spatial dimensions of "production space, living space, ecological space" in the face of internal and external disturbances. The interactions between these three types of spaces can be represented in Fig 1. The development level of each production-living-ecological spaces reflects, to some extent, the city's level of economic development, social coordination, ecological environment development, and infrastructure building. The primary role of urban production space is to fulfil the economic demands of city dwellers; it is the place where industrial activities are concentrated and has the function of producing material materials and services in the city. As a result, production economic indicators may be used to measure the development of production space resilience. The primary role of urban living space is to fulfil the material and spiritual requirements of people, and it is the area for human social activities, containing the livability and high quality of life functions for urban inhabitants.

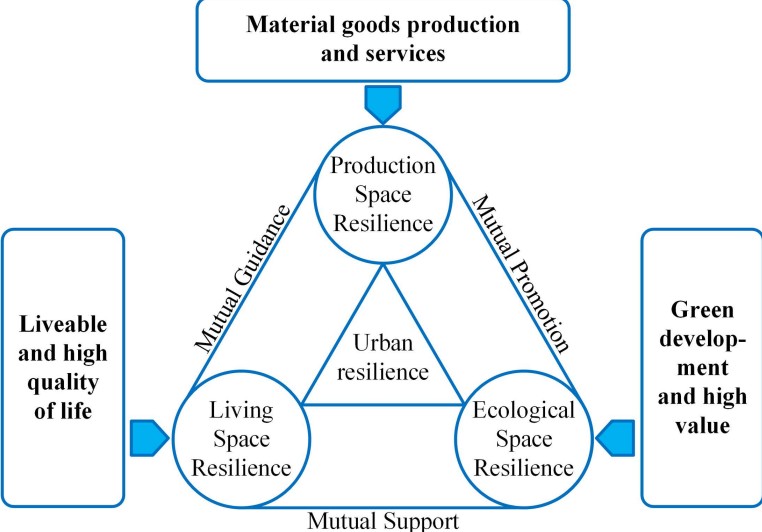

**Fig 1. The conceptual map of production-living-ecological spatial resilience.**

As a result, social, demographic, and infrastructural indices are used to assess the resilience of urban living spaces. The primary role of urban ecological space is to fulfil the resource demands of urban people and to safeguard the environment, which is a significant manifestation of the city's green development and high value. It is mostly measured using measures like urban greening, pollution and treatment, and geological catastrophe and protection.

As a result, this study creates a complete spatial evaluation index system for the resilience assessment of counties and districts from many areas such as economics, society, environment, and infrastructure based on the three geographical scales of production-living-ecological spaces. At the micro-scale, eight counties in the middle and lower reaches of the Three Gorges Reservoir Area in China's less developed regions are used as research objects to fill the urban resilience research gap; the evaluation indexes are analyzed and calculated by geographic probes and other methods to obtain the ranking of impact factors, in order to propose targeted resilience improvement strategies, which are of great significance to the sustainable development of the Three Gorges Reservoir Area in China.

## 2. Materials and methods

### 2.1 Study area

The Three Gorges Reservoir Area in China refers to a few administrative regions in China that were submerged as a result of the construction of the Three Gorges project. These regions had migration responsibilities. The area that contains the reservoir is situated in the combination of the Sichuan Basin and the middle and lower reaches of the Yangtze River Plain. It stretches across the mountainous valley of the central E and the valley of the eastern Sichuan Ridge, with the Daba Mountains to the north and the Sichuan-E plateau to the south [30]. The overall landmass spans around 58,000 km². It is made up of 22 districts and counties in Chongqing as well as four districts and counties in the province of Hubei. The Yangtze River Economic Belt connects the middle reaches of the Yangtze River urban agglomeration and the Chengdu-Chongqing urban agglomeration. The Three Gorges Reservoir Area is located in the Yangtze River Economic Belt, and the development of the counties and districts in the Three Gorges Reservoir Area is particularly important as the Yangtze River Economic Belt strategy moves forward [26]. However, because the Three Gorges Reservoir Area is a new geographic area formed by relocated migrants, the conflicts left over from the migration period combined with its special geographical environment and development model have led to the fading of the original urban characteristics of the counties, the intensity of social conflicts, a fragile ecological environment, uneven urban and rural development, and serious geological problems, putting the area at a disadvantage in subsequent development. Therefore, conducting research on the production-living-ecological spatial resilience in the reservoir region and gaining an understanding of its resilience-driven mechanism may give both theoretical and guiding relevance for the purpose of further boosting the strategic growth of the Yangtze River Economic Belt [31,32].

Yiling District, Zigui County, Xingshan County, and Badong County in Hubei Province are studied, as are Wanzhou District, Yunyang County, Fengjie County, and Wushan County in Chongqing City (Fig 2). These districts and counties are located in the first section of the reservoir in Hubei Province and the middle section of the reservoir in Chongqing City, respectively. The era of the Three Gorges Reservoir Area is defined by the construction of the Three Gorges Water Conservancy Hub Project. The preparation period is defined as the "pre-Three Gorges period", The project officially commenced on December 14, 1994. The main body of the dam was completed on May 20, 2006, and the entire project was fully finished in October 2008. The overall completion acceptance was carried out on November 1, 2020. The construction period is defined as the "migrant new city period", and the completion period is defined as the "post-Three Gorges period" [33]. This study focuses on the time period following the completion of the Three Gorges Project and considers the years 2010, 2015, and 2020.

### 2.2 Data source

Spatial data, DEM (Digital Elevation Model) data, and accessible government information are all included in the study data. The spatial data are taken from remote sensing monitoring data of land use/cover for 2010, 2015, and 2020 with a spatial resolution of 30m provided by the Resource and Environment Science Data Center of the Chinese Academy of

Sciences (http://www.resdc.cn/). The DEM (Digital Elevation Model) data are taken from the Geospatial Data Cloud Platform's ASTER GDEM V2 global digital elevation data product (http://www.gscloud.cn/), with a spatial resolution of 30m. The open government statistics are obtained from the <Statistical Yearbook of County Construction in China>,<Government Work Report>,<Annual Report on Environmental Quality>,<Statistical Yearbook>,<Statistical Bulletin of National Economic and Social Development>,<Geological Disaster Risk Assessment Report>,<Annual Report on Environmental Quality>,<Soil and Water Conservation Bulletin>, as well as the<12th Five-Year Plan>,<13th Five-Year Plan> and <14th Five-Year Plan> of each government department. For some missing data, the mean filling method and the growth rate filling method are adopted to fill in the missing data to ensure the smooth implementation of the study. The mean imputation method involves calculating the spatial mean of data points, and the proportion of missing data in this study is less than 5%.

## 2.3 Evaluation system construction

**2.3.1 Tenacity evaluation ideas.** The general research concept of this article is presented in Fig 3: (1) The urbanization backdrop gives a topic idea for this thesis, and prior research provides the underlying theory and technology. (2) A resilience evaluation index system applicable to the Three Gorges Reservoir Area is built based on the special ecological environment and economic conditions of the reservoir area, and indexes with a high coefficient of variation

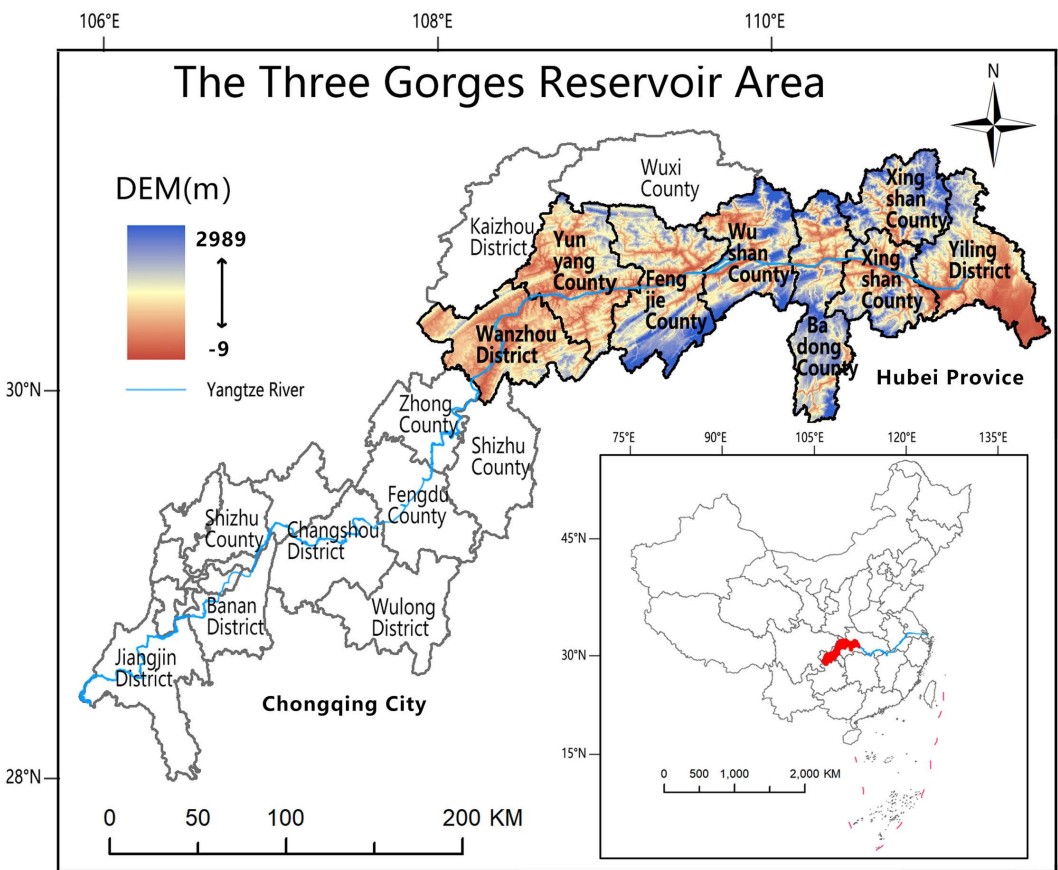

**Fig 2. Topographic map of the study area.** Source: Created by the author using a base map from the National Catalogue Service for Geographic Information (https://www.webmap.cn/), with no modification to its boundaries.

are chosen to build the driving force evaluation system. (3) Using the comprehensive evaluation method, we calculate the weights of each indicator and the resilience index of each subspace to calculate the comprehensive resilience index of counties and districts and then use the Geographic detector to calculate and analyze the driving forces, providing theoretical support for the resilience enhancement recommendations in the subsequent paper.

**2.3.2 Resilience indicator system construction.** Economic output efficiency and industrial structure development are two of the functions that are included in production space, which is the geographic component of a city or town's economic and industrial output efficiency. As a result, the resilience of production space is equivalent to the ability of urban economies and industries to recover from disruptions. To evaluate the production resilience, nine indicators have

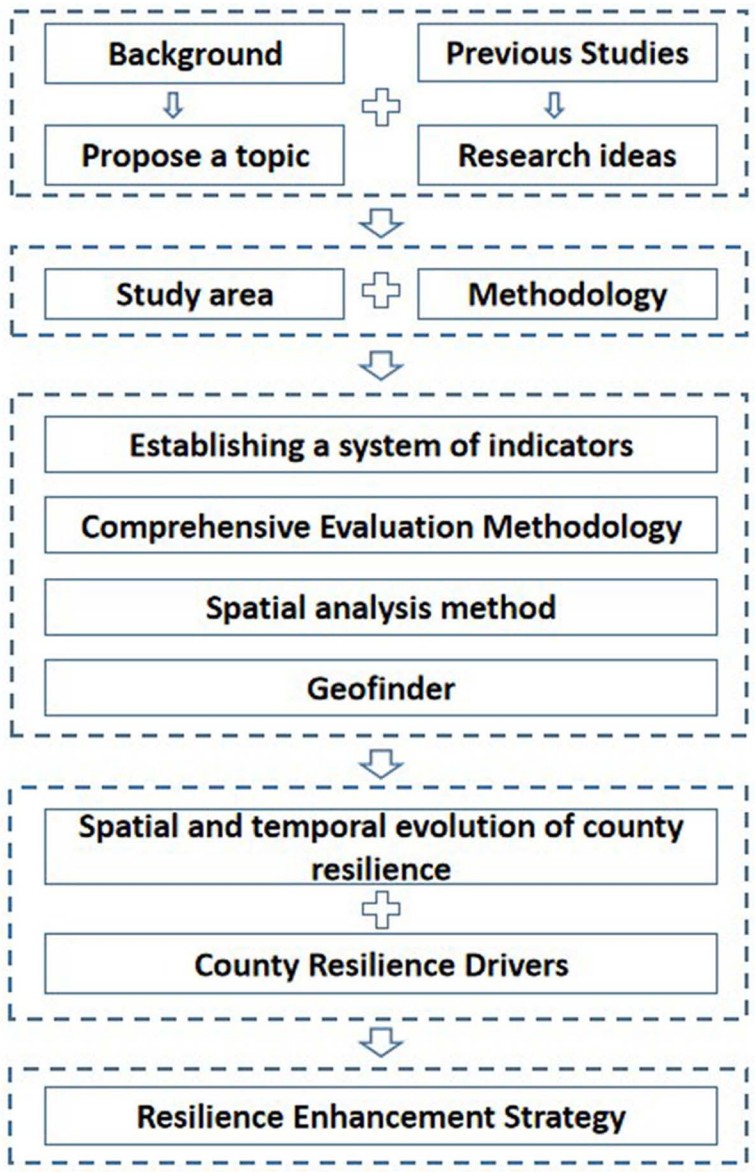

**Fig 3. Flow chart of this paper's research.**

been chosen. Living space is the spatial scope for humans to carry out daily activities, and it has the area to meet both the material life and the spiritual civilised life of residents. As a result, living space resilience places an emphasis on the living conditions and social growth of residents. Nop and Thornton's 2019 study found that cities with higher levels of social capital have higher levels of resilience [30]. In order to evaluate the resilience of living space, fourteen indicators have been chosen. A significant portion of a city is comprised of urban ecological space. Ecological space places an emphasis on the environmentally responsible growth of the city, which is primarily expressed in ecological conservation, risk prevention and control, ecological management, etc. A healthy and stable ecological environment contributes to the assurance of the city's continued growth in a sustainable direction. For the purpose of determining the resilience of ecological space, seven indicators are chosen. In conclusion, through a comprehensive literature review and expert consultation, we carefully considered the unique characteristics of the Three Gorges Reservoir Area and selected 30 indicators that possess availability, representativeness, and comprehensiveness. These indicators were used to construct a resilience assessment index system for production-living-ecological spaces, with the aim of evaluating the resilience of eight counties in China's Three Gorges Reservoir Area. (Table 1).

**2.3.3 Driving force indicator identification.** According to the previous analysis, there is significant spatial and temporal variation in urban resilience characteristics in the middle and lower reaches of the Three Gorges Reservoir Area in China, and it is influenced by a variety of factors, including production spatial factors, living spatial factors, and ecological spatial factors. In this article, we apply the evaluation index method to analyze the impact factors, according to the existing literature [34–36] and filter the indicators with a high coefficient of variation. Three aspects are chosen from the production space: effectiveness, efficiency, and size. Three factors are chosen from the living space: consumption, transit, and information. Three aspects are chosen from the ecological space: greening, sewage, and land disaster. A total of nine indicators are selected (Table 2): fixed asset investment per unit area, fixed asset investment per unit area (X1), GDP per capita (X2), total foreign trade import and export per unit area (X3), per capita retail sales of social consumer goods (X4), road area per capita (X5), Internet penetration rate (X6), forest coverage rate (X7), sewage treatment standard rate (X8), and density of monitoring points for early warning group prevention of disaster (X9). The data of each index are classified and transformed into type data using SPSS. The type data are transferred into Geodetector, and the drivers are analyzed based on Factor detector and Interaction detector measurements.

## 2.4 Comprehensive evaluation method

**2.4.1 Data standardization processing.** The data metrics employed in this work are heterogeneous, with a huge quantity of data and inconsistent data units, making direct computations impossible. To improve the accuracy of the study's results, the impact of differing unit magnitudes on the results must be eliminated [37]. As a result, after removing outliers, the processed data must be further standardized to transform the various unit values into relative values of the same unit. The deviation standardization approach is used in this study to linearly modify the original indicator data such that the results are mapped between 0 and 1. Meanwhile, the positive and negative meanings of the indicator are highlighted (the positive nature of the indicator indicates that the higher its value represents the better indicator meaning; the negative nature of the indicator indicates that the lower its value represents the better indicator meaning). The following are the specific data pre-processing methods:

$$V_{ij} = \frac{Y_{ij} - Min(Y_j)}{Max\ (Y_j) - Min(Y_j)}$$

(1)

$$V_{ij} = \frac{Min\ (Y_j) - Y_{ij}}{Max\ (Y_j) - Min(Y_j)}$$

(2)

**Table 1.** Resilience evaluation index system based on production-living-ecological spaces.

| Target layer | Criterion layer | Indicator layer | Weights | Nature of indicators |
|---|---|---|---|---|
| **Comprehensive toughness** | Production space resilience | GDP per capita (C1) | 0.0382 | + |
| | | The proportion of tertiary industry (C2) | 0.0143 | + |
| | | Percentage of public revenue (C3) | 0.0198 | + |
| | | Percentage of total tourism revenue (C4) | 0.0315 | + |
| | | Percentage of total industrial output value (C5) | 0.0504 | + |
| | | Agriculture, forestry, animal husbandry and fishery services as a percentage (C6) | 0.0194 | + |
| | | Total foreign trade import and export ratio (C7) | 0.1567 | + |
| | | Deposits in financial institutions per capita (C8) | 0.0359 | + |
| | | Unit area fixed asset investment (C9) | 0.0549 | + |
| | Living space resilience | Road area per capita (C10) | 0.0739 | + |
| | | Density of drainage pipes in built-up areas (C11) | 0.0427 | + |
| | | Disposable income per capita (C12) | 0.0315 | + |
| | | Retail sales of consumer goods per capita (C13) | 0.0485 | + |
| | | Number of beds per 10,000 people (C14) | 0.0283 | + |
| | | Percentage of students enrolled (C15) | 0.0412 | + |
| | | Urban registered unemployment rate (C16) | 0.0182 | − |
| | | Housing area per capita (C17) | 0.0432 | + |
| | | Water supply per capita (C18) | 0.0126 | − |
| | | Gas penetration rate (C19) | 0.0251 | + |
| | | Internet penetration rate (C20) | 0.0467 | + |
| | | Percentage of population over 60 years old (C21) | 0.0209 | − |
| | | Population density (C22) | 0.0148 | − |
| | | Urbanization level (C23) | 0.0287 | + |
| | Ecological space resilience | Wastewater treatment compliance rate (C24) | 0.0115 | + |
| | | Forest cover (C25) | 0.0265 | + |
| | | Air Quality Index (C26) | 0.0087 | + |
| | | Greening coverage of built-up areas (C27) | 0.0090 | + |
| | | Green space per capita (C28) | 0.0149 | + |
| | | Proportion of area in high geological hazard-prone areas (C29) | 0.0131 | − |
| | | Density of monitoring points for early warning group prevention of earthquakes (C30) | 0.0189 | + |

For the above equation, $V_{ij}$ is the $j$ normalized value of the $i$ spatial function; $Y_{ij}$ is the original value of the $j$ indicator of the $i$ spatial function; $Min$ is the minimum value of the spatial function; $Max$ is the maximum value of the spatial function, when the indicator is positive, the data pre-processing formula is Eq (1), and when the indicator is negative, the data processing formula is Eq (2).

**2.4.2 Calculating the toughness index.** We calculate the urban resilience indices in this paper using multiple indicator weights, and to avoid the interference of human factors and the problem of overlapping information between indicators, the entropy weight method is used [38,39]. This approach is commonly used for a thorough evaluation of various indicators and can properly judge data dispersion. The wider the dispersion of indicators, the more effective

**Table 2. Indicator system of resilience drivers for towns in the Three Gorges Reservoir Area in the context of three living spaces.**

| Objective level | Guideline level | Indicator level | Indicator meaning | Calculation method |
|---|---|---|---|---|
| **Integrated resilience drivers** | Production space resilience driver | Fixed asset investment per unit area (X1) | Production efficiency | Amount of fixed asset investment/ urban land area |
| | | GDP per capita (X2) | Economic efficiency | Local GDP/ Population |
| | | Total foreign trade import and export per unit area (X3) | Scale of production | Total foreign trade import/ export/ area of urban land |
| | Living space resilience driver | Per capita retail sales of social consumer goods (X4) | Consumption level | Total social retail consumption/ number of urban population |
| | | Road area per capita (X5) | Transportation level | Existing road area/ town population |
| | | Internet penetration rate (X6) | Information and communication | Number of Internet users/ total number of users*100% |
| | Ecological space resilience driver | Forest coverage rate (X7) | Greening level | Forest coverage area/ town land area*100% |
| | | Sewage treatment standard rate (X8) | Sewage disposal capacity | Amount of sewage treated to standard/ amount of sewage produced*100% |
| | | Density of monitoring points for early warning group prevention of earthquakes (X9) | Earthquake prevention | Number of monitoring points for early warning and group prevention of earthquakes/ area of urban land |

information, the greater the entropy value and the bigger the weight; when the indicators have the same data, there is no useful information, and the weight is 0. The steps for calculating are as follows:

The indicator weights are calculated with the formula:

$$W_{ij} = \frac{V_{ij}}{\sum_i \sum_j V_{ij}} \tag{3}$$

Calculate the entropy value of the $j$th indicator with the formula:

$$e_j = -\frac{\left[\sum_{i=1}^n W_{ij} \ln\left(W_{ij}\right)\right]}{\ln n} \tag{4}$$

The effective information value of the $j$th indicator is calculated by the formula:

$$k_j = 1 - e_j \tag{5}$$

Calculate the weight of the $j$th indicator with the formula

$$e_{ij} = \frac{k_j}{\sum_j^n k_j} \tag{6}$$

Calculate the weights of each spatial system with the formula

$$e_{ij} = \sum_{j=1}^m e_{ij} \tag{7}$$

Calculate the resilience evaluation index of each spatial system with the formula,

$$RES_a^s = \sum_{j=1}^{m} (k_j V_{ij})$$

(8)

Calculate the composite assessment value of toughness with the formula:

$$RES = \sum_{j=1}^{m} (RES_a^s)_j k_j$$

(9)

Where $W_{ij}$ is the indicator weight, $V_{ij}$ is the value calculated by [Eqs (1)](1) and [(2)](2), $e_j$ is the entropy value of the $j$th item, $k_j$ is the effective information value of the $j$th item, $e_{ij}$ is the weight of the $j$th indicator, $e_{jj}$ is the weight of the sub-spatial system, $RES_a^s$ is the resilience evaluation index of the $a$th year s system, $RES$ is the comprehensive evaluation index of the urban resilience evaluation, $n$ is the number of indicators, $m$ is the number of indicators contained in each spatial system.

## 2.5 Spatial analysis method

In ArcGIS software, the Jenks natural breakpoint method can use its own characteristics to partition the data into multiple independent grades based on a continuous value space. It is commonly utilized in different visualization analyses because it successfully avoids the smoothing effect induced by the fixed breakpoint technique and makes the visualization findings more intuitive [40]. After grading, spatial autocorrelation is used to analyze the overall spatial correlation, difference and accumulation, and Moran's I value can be used to determine whether there is an anomaly in the space. When $0 <$ Moran's I value $<= 1$, the space shows a positive correlation, and the more clustered the distribution, the stronger the correlation; when $-1 = <$ Moran's I value $< 0$, the space shows a negative correlation, the more discrete the distribution, the stronger the correlation; when Moran's I value $= 0$, the space is randomly distributed. The Moran's I index formula is:

$$I = \frac{\sum_i^n \sum_j^n W_{ij} (A_i - \overline{A})(A_j - \overline{A}) n}{\sum_i^n \sum_j^n W_{ij}(A_i - \overline{A})^2}$$

(10)

Where $W_{ij}$ is the matrix of spatial weights of $i$ and $j$, given in terms of distance, $A_i$ and $A_j$ signify the toughness values of city $i$ and city $j$ respectively, $\overline{A}$ is the mean value of toughness, and $n$ is the number of studies.

## 2.6 Exploring resilience drivers

The Geodetector is a geographic detection model that employs statistics to investigate the spatial differentiation of geographical entities influencing elements in their spatial variability. The model has four detectors: Factor detector, Interaction detector, Risk detector, and Ecological detector [41,22]. Geographic detectors are employed in this article to explore the drivers of resilience in the middle and downstream counties of China's Three Gorges Reservoir Area. Factor detectors were used to detect the strength of the drivers on the county's resilience, and interaction detectors were used to find the explanatory power of the factors following interactions between the drivers [42,43]. Table 3 displays the interaction kinds of the interaction detectors. The dependent variable is the resilience index of the Three Gorges Reservoir Area in China in 2011, 2015, and 2020, and the independent variables are continuous-type variables classified after discrete processing of the evaluation indicators, and the strength of the independent variables' influence on the dependent variable is indicated by, and the explanatory power is indicated by, with larger values indicating stronger influence and smaller values indicating weaker influence [44].

**Table 3. Interaction types of interaction probes.**

| Judgment basis | Interaction |
|---|---|
| $q(X1 \cap X2) < min[q(X1), q(X2)]$ | Nonlinear attenuation |
| $min[q(X1), q(X2)] < q(X1 \cap X2) < max[q(X1 \cap X2)]$ | Single-factor nonlinear weakening |
| $q(X1 \cap X2) > max[q(X1), q(X2)]$ | Two-factor enhancement |
| $q(X1 \cap X2) = q(X1) + q(X2)$ | Mutually independent |
| $q(X1 \cap X2) > q(X1) + q(X2)$ | Non-linear enhancement |

**Table 4. Tenacity index of eight counties in the middle and lower reaches of the Three Gorges Reservoir Area in China in 2010.**

| Area | Production Space | Living Space | Ecological Space | Comprehensive status |
|---|---|---|---|---|
| **Yiling District** | 0.0728 | 0.0914 | 0.0479 | 0.2121 |
| **Xingshan County** | 0.0956 | 0.0899 | 0.0556 | 0.2411 |
| **Zigui County** | 0.0576 | 0.0733 | 0.0513 | 0.1823 |
| **Badong County** | 0.0506 | 0.0766 | 0.0496 | 0.1767 |
| **Wanzhou District** | 0.0628 | 0.1936 | 0.0377 | 0.2941 |
| **Yunyang county** | 0.0413 | 0.0947 | 0.0477 | 0.1836 |
| **Fengjie County** | 0.0399 | 0.0971 | 0.0385 | 0.1755 |
| **Wushan County** | 0.0401 | 0.0989 | 0.0374 | 0.1764 |

## 3 Results

### 3.1 Spatial and temporal variation in urban resilience

The comprehensive evaluation method was used to calculate each space and county's comprehensive toughness of eight counties in the middle and lower reaches of the Three Gorges Reservoir Area in China in 2010, 2015, and 2020, and the results of each county were processed and analyzed using ArcGIS software to obtain the spatial distribution of toughness of counties in the middle and lower reaches of the Three Gorges Reservoir Area in China in 2010, 2015, and 2020, respectively (Tables 4–6; Figs 4–6).

**3.1.1 Spatial and temporal evolutionary characteristics of the county's comprehensive resilience characteristics.** In terms of time, the resilience of the middle and downstream counties in China's Three Gorges Reservoir region shows a progressive rising tendency, with each county achieving its peak in 2020. Each county in the reservoir area had the lowest resilience in 2010, the year when water conservancy projects in the Three Gorges Reservoir Area were initially completed, with Fengjie, Wushan, and Badong counties having relatively low resilience, Yunyang and Zigui counties having medium resilience, Xingshan, Yiling, and Wanzhou counties having high resilience, and Yiling and Wanzhou districts being high toughness areas. With the successful completion of the first five-year plan of  in 2015, the counties' production and living gradually improved, and the counties' toughness increased significantly, with Wushan County's toughness improving faster, from a lower toughness area to a medium toughness area, and the toughness of Xingshan County, Yiling District, and Wanzhou District remaining steady at a high level. By 2020, the toughness of counties and districts had been consistently enhanced as a result of long-term positive growth, with Zigui County having the quickest increase and being a higher toughness region together with Xingshan County, Yiling District, and Wanzhou District.

The resilience level of counties in the middle and lower reaches of China's Three Gorges Reservoir region exhibits a pattern of "high on both sides and low in the middle" from a geographical perspective. Low resilience counties are mostly located in counties surrounding the main urban areas of Hubei Province and Chongqing City, where production capacity is low, living space infrastructure is inadequate, and the ecological space environment is unstable. High toughness is located

**Table 5. Tenacity index of eight counties in the middle and lower reaches of the Three Gorges Reservoir Area in China in 2015.**

| Area | Production Space | Living Space | Ecological Space | Comprehensive status |
|---|---|---|---|---|
| Yiling District | 0.1683 | 0.1807 | 0.0579 | 0.4068 |
| Xingshan County | 0.1455 | 0.2043 | 0.0603 | 0.4101 |
| Zigui County | 0.1206 | 0.1438 | 0.0577 | 0.3220 |
| Badong County | 0.0813 | 0.0944 | 0.0443 | 0.2200 |
| Wanzhou District | 0.1438 | 0.2955 | 0.0525 | 0.4917 |
| Yunyang county | 0.0853 | 0.1407 | 0.0611 | 0.2871 |
| Fengjie County | 0.0696 | 0.1618 | 0.0420 | 0.2735 |
| Wushan County | 0.0828 | 0.1544 | 0.0632 | 0.3004 |

**Table 6. Tenacity index of eight counties in the middle and lower reaches of the Three Gorges Reservoir Area in China in 2020.**

| Area | Production Space | Living Space | Ecological Space | Comprehensive status |
|---|---|---|---|---|
| Yiling District | 0.1543 | 0.1874 | 0.0738 | 0.4156 |
| Xingshan County | 0.2642 | 0.2340 | 0.0753 | 0.5735 |
| Zigui County | 0.1840 | 0.1792 | 0.0772 | 0.4404 |
| Badong County | 0.0762 | 0.1808 | 0.0770 | 0.3340 |
| Wanzhou District | 0.1083 | 0.3387 | 0.0625 | 0.5095 |
| Yunyang county | 0.0923 | 0.2026 | 0.0604 | 0.3553 |
| Fengjie County | 0.0892 | 0.2337 | 0.0658 | 0.3887 |
| Wushan County | 0.0983 | 0.2123 | 0.0672 | 0.3778 |

mostly around the city's main urban regions, where the production function is more steady due to the radiation caused by the main urban areas, the living space infrastructure is more comprehensive, and the ecological space environment is better regulated. According to the spatial analysis method, 2010, 2015 research area resilience Moran's I values are positive, indicating that the urban resilience in the study area has positive spatial correlation characteristics and the urban resilience near the region has spatial clustering phenomenon, and the values decreased from 2010 to 2015, indicating that the county spatial correlation gradually weakened; 2020 research area Moran's I has negative values, suggesting that there is no major agglomeration at the county level, and the presence of oscillations in Moran's I suggests that spatial accumulation across districts and counties is unstable.

**3.1.2 Spatial and temporal evolution characteristics of urban resilience in each subspace.** The spatial resilience of production in the middle and lower portions of China's Three Gorges Reservoir region exhibits three tendencies throughout time: "increasing and then decreasing", "continuously increasing", and "decreasing and then increasing". The spatial resilience of production in Yiling District, Badong County, and Wanzhou District, in particular, increases and then decreases, while the spatial resilience of production in Xingshan County, Zigui County, Fengjie County, and Wushan County continues to increase, and that of production in Yunyang County decreases and then increases. In 2010, Wanzhou District, Xingshan County, and Yiling District production space toughness showed a high level; Badong County and Zigui County showed a medium level; Yunyang County, Fengjie County, and Wushan County showed a low level; the overall production space toughness level was consistent with the county comprehensive toughness level. In 2015, Yiling District, Xingshan County, Zigui County, and Wanzhou District showed a high level; Yunyang County exhibited a medium level; Fengjie County, Wushan County, and Badong County presented a low level. In 2020, Yiling District, Xingshan County, and Zigui County presented a high level, Wanzhou presented a medium level, and Yunyang County, Fengjie County, Wushan County, and Badong County presented a low level. The production space toughness level of each county fluctuates less, and the spatial variability is not significant.

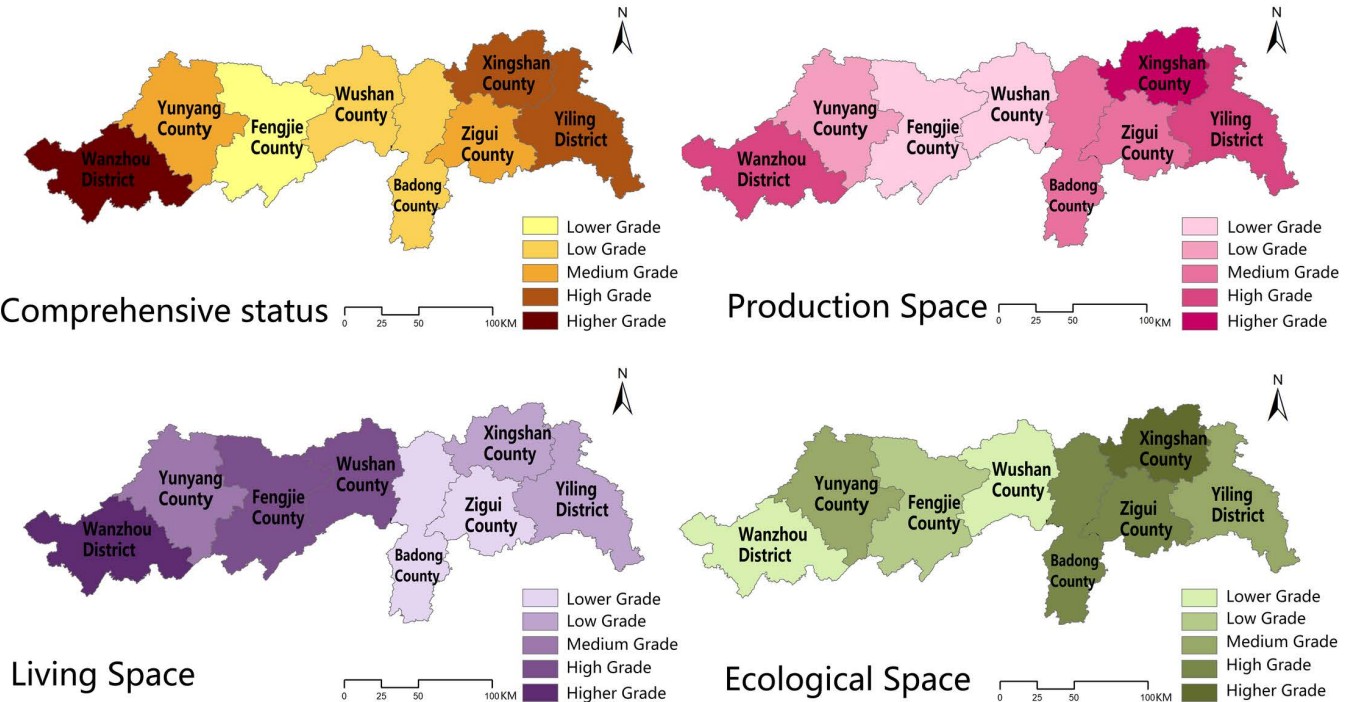

**Fig 4. Spatial distribution of resilience in the middle and lower reaches of the Three Gorges Reservoir Area counties in China, 2010.** Source: Created by the author using a base map from the National Catalogue Service for Geographic Information (https://www.webmap.cn/), with no modification to its boundaries.

The living space toughness is the highest of the three sub-spatial toughness, and all eight counties and districts exhibit a progressively growing trend of living space toughness, reaching the maximum in 2020. In particular, in 2010, Wanzhou District, Fengjie County, and Wushan County were at a high level, Yunyang County was at a medium level, and Badong County, Zigui County, Xingshan County, and Yiling District were at a low level; in 2015, Wanzhou District, Xingshan County, and Yiling District were at a high level, Fengjie County and Wushan County were at a medium level, and Yunyang County, Zigui County, and Badong county were at a low level; Wanzhou District, Fengjie County, and Xingshan County had high resilience in 2020, Yunyang County and Zigui County had medium resilience, while Badong County, Zigui County, and Yiling District had poor resilience. The change in living space fluctuates a lot, and its form is comparable to the change in comprehensive resilience level, with significant regional differences.

The ecological spatial toughness displays a fluctuating increasing trend and steadily increases with time, peaking in 2020. Except for Badong and Yunyang counties, which exhibit a lowering and then increasing trend in ecological spatial resilience, all other counties and districts show a progressive increase. In 2010, Xingshan, Zigui, and Badong counties had high ecological spatial resilience; Yiling District and Yunyang County had medium resilience; Wanzhou District, Fengjie County, and Wushan County had low resilience. In 2015, Wushan, Yunyang, and Xingshan counties had high ecological spatial resilience, Zigui and Yiling districts had medium resilience, and Wanzhou District, Fengjie County and Badong County had low resilience. In 2020, Xingshan County, Yiling District, Zigui County, and Badong County had high toughness; Fengjie County and Wushan County had medium toughness; Wanzhou District and Yunyang County had low toughness. The ecological spatial toughness levels changed less, and there was no substantial geographic fluctuation.

The spatial analysis approach examines the values for subspace spatial accumulation, production space toughness, and living space toughness. Moran's I index is positive, indicating a positive correlation between production space and

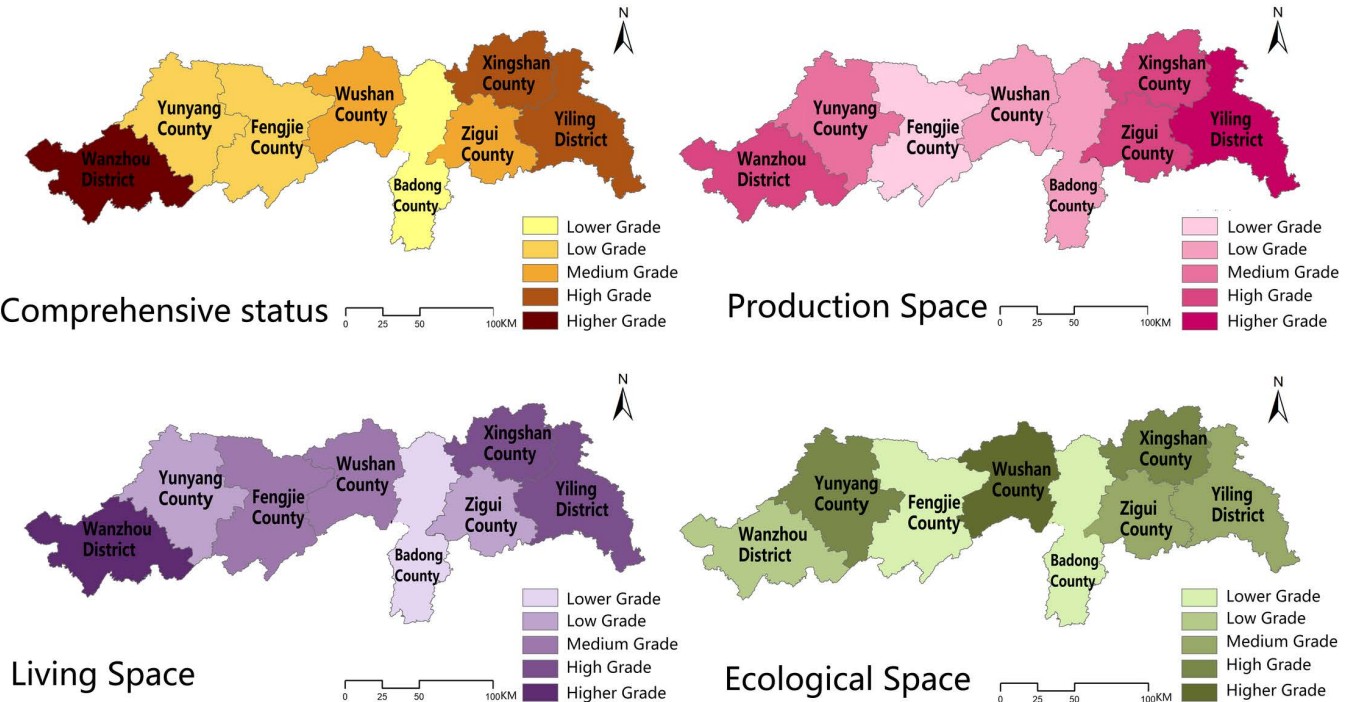

**Fig 5. Spatial distribution of resilience in the middle and lower reaches of the Three Gorges Reservoir Area counties in China, 2015.** Source: Created by the author using a base map from the National Catalogue Service for Geographic Information (https://www.webmap.cn/), with no modification to its boundaries.

living space, and living space is greater, indicating that the living space system in the study area is more important. Moran's I for ecological spatial resilience drops from positive to negative and then increases to positive values, showing that the county's ecological spatial accumulation is very unstable.

### 3.2 Resilience drivers

**3.2.1 Driving force results.** The factor and interaction detectors were used to determine the values of the explanatory strength of the urban resilience driver factor and the interaction of the two indicators. The higher the q-value, the larger the explanatory power of the indicator's effect on the findings, and the results of its factor detector q-values for 2010, 2015, and 2020 are presented below (Table 7).

According to the geographic detector results, the q values are inside the 0−1 range. The values are classified as I primary driver (0.8−1), II important driver (0.6–0.8), III more important driver (0.4–0.6), and IV secondary driver (0–0.4), and their relevance is rated as I primary driver＞II important driver＞III more important driver＞IV secondary driver. Economic Efficiency, Production Efficiency, Consumption Level, and Transportation Level drivers with q-values of more than 0.8 were more significant for the resilience of the counties in the middle and lower reaches of the Three Gorges Reservoir region in 2010 and were the key drivers of level I. Information and Communication, Greening Level, and Disaster Prevention drivers with q-values more than 0.6 and less than 0.8 were the second most effective for the middle and lower portions of the Three Gorges Reservoir Area and were major drivers of level II. The q value of the Sewage Disposal Capacity driver was larger than 0.4 but less than 0.6, indicating that it was a more important driver for level III. The q value of the Production Efficiency driving force was less than 0.4, making it the smallest level driving factor and the secondary driving component of level IV. In 2015, the driving force q value of Economic Efficiency, Production Efficiency, Consumption Level, and Sewage Disposal Capacity was greater than 0.8, which was more significant for the resilience driving of the middle

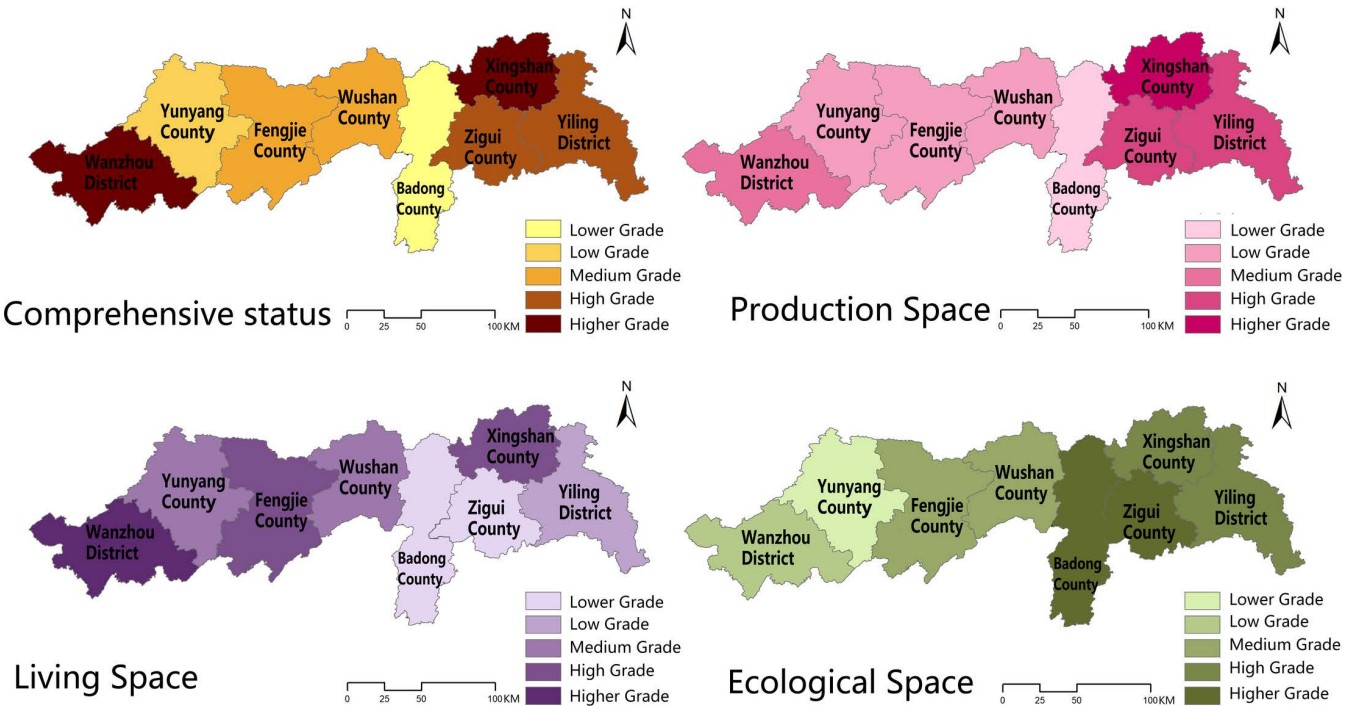

**Fig 6. Spatial distribution of resilience in the middle and lower reaches of the Three Gorges Reservoir Area counties in China, 2020.** Source: Created by the author using a base map from the National Catalogue Service for Geographic Information (https://www.webmap.cn/), with no modification to its boundaries.

**Table 7. Factor detection probing q-values for the Three Gorges Reservoir Area, 2010, 2015, 2020.**

| Year | X1 | X2 | X3 | X4 | X5 | X6 | X7 | X8 | X9 |
|------|------|------|------|------|------|------|------|------|------|
| **2010** | 0.3247 | 0.9504 | 0.9520 | 0.9194 | 0.9211 | 0.6865 | 0.7934 | 0.4297 | 0.6367 |
| **2015** | 0.6709 | 0.8738 | 0.8691 | 0.9230 | 0.6792 | 0.7961 | 0.4074 | 0.8804 | 0.6362 |
| **2020** | 0.5235 | 0.8996 | 0.9244 | 0.7727 | 0.7682 | 0.5554 | 0.6922 | 0.5136 | 0.7638 |

and downstream counties in the Three Gorges Reservoir Area, and was the primary driving factor of level I. The driving power was larger than 0.6, less than 0.8 for Production Efficiency, Traffic Level, Information and Communication, and Disaster Prevention, and the driving impact was only second to level I, which was an important driving component of level II. The q-value of the greening level was larger than 0.4 and less than 0. 6, the smallest driving force level in 2015 and the more important driving element of level III. In 2020, the Economic Efficiency, Production Efficiency driving force q-value was larger than 0.8, more significant for the urban resilience driver of the middle and lower ranges of the Three Gorges Reservoir region, making it the principal driving factor of level I. Consumption Level, Transportation Level, Greening Level, and Disaster Prevention drivers with q-values larger than 0.6 and less than 0.8 were less effective than I for driving the counties in the middle and lower reaches of the Three Gorges Reservoir region but were major drivers for II. The q-values for Production Efficiency, Information and Communication, and Sewage Disposal Capacity were larger than 0.4 and less than 0.6, indicating that they were more important driving variables for Class III.

Economic Efficiency, Production Efficiency, and Consumption Level are the most important drivers of resilience in the Three Gorges Reservoir Area's middle and lower reaches. From 2010 to 2020, the resilience of the Three Gorges Reservoir Area's middle and lower reaches was primarily driven by the production space driver, followed by the living space driver and the ecological space driver.

**3.2.2 Driving force interaction results.** The interaction detection effect analysis was performed using the Factor detector. When the analysis indicators interacted with each other two by two, the effect results were all greater than the individual indicators, and the interactions were mostly two-factor enhancement, with some non-linear enhancement, indicating that the resilience of the county and region was subject to multiple factor interactions. Among these, the impact of two-factor enhancement outperforms the effect of each individual factor, the effect of non-linear enhancement outperforms the sum of their individual effects, and the effect of non-linear enhancement outperforms the effect of two-factor enhancement. The county-area space and subspace two-indicator interaction findings for 2010, 2015, and 2020 are provided below (Figs 7–9).

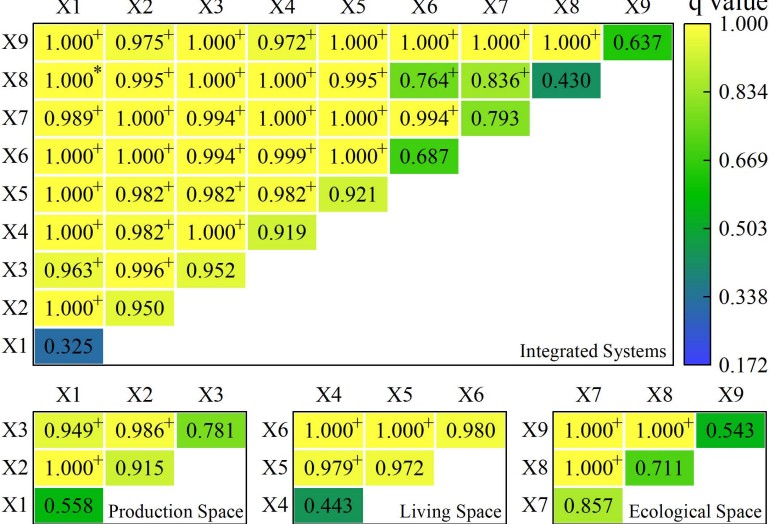

**Fig 7. Results of the interaction between two indicators in town space and subspace in 2010.** Note: * represents a non-linear enhancement; + represents a two-factor enhancement; no markers represent no significant difference.

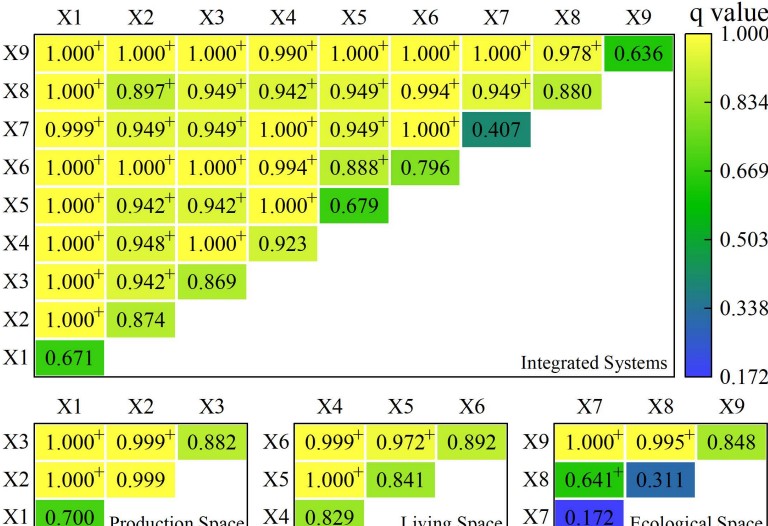

**Fig 8. Results of the interaction between two indicators in town space and subspace in 2015.** Note: + represents a two-factor enhancement; no markers represent no significant difference.

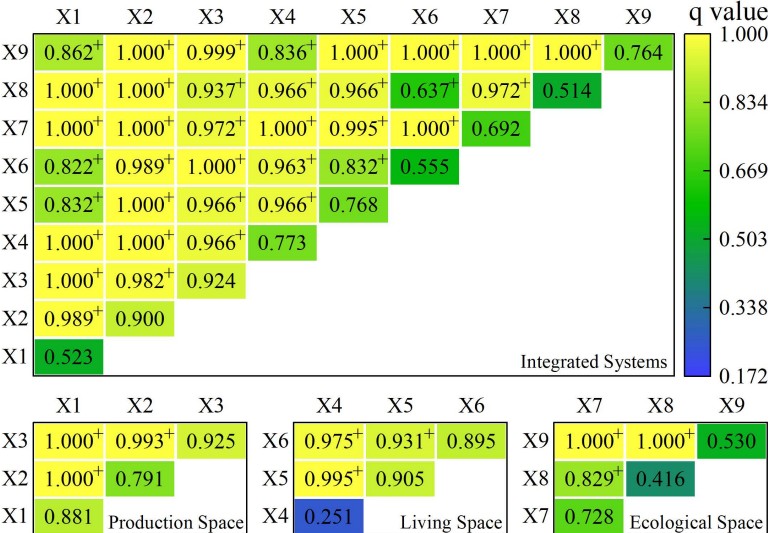

**Fig 9. Results of the interaction between two indicators in town space and subspace in 2020.** Note: + represents a two-factor enhancement; no markers represent no significant difference.

## 4 Discussion

As a city is a complex multi-role system, investigating urban resilience and increasing cities' ability to withstand shocks and quickly restore stability is the foundation for high-quality urban growth and is a popular issue in contemporary urban research [45,46]. Understanding the spatial and temporal differentiation of resilience and identifying drivers can provide a favourable theoretical basis and technical guidance for improving urban resilience in the context of the still fragile environment of the Three Gorges Reservoir Area counties in China. Although research on urban resilience assessment is rather mature, there are few studies on the microscopic size of urban resilience in less-developed regions, as well as few studies on the evaluation system based on the spatial dimension of the production-living-ecological spaces. In this paper, we take the first and middle sections of the Three Gorges Reservoir Area in China as the research object, filling a gap in urban resilience research of less-developed areas at the micro-scale, constructing a resilience evaluation system and a driver evaluation system based on the spatial dimension of the production-living-ecological spaces, calculating and analyzing the spatial and temporal evolution of resilience and the driver of the Three Gorges Reservoir Area. This study provides theoretical support for the development of resilient cities in order to establish a more scientific and acceptable development plan for city construction.

The study objects for this work are limited to only a few counties in the first and middle parts of the Three Gorges Reservoir Area in China, and it does not make any horizontal comparisons with other cities in China. Second, we only analyzed the scenario regarding resilience during the years 2010–2020 without simulating the growth of the future, which may result in less precise placement in following building planning. These are two of the limitations of this paper. In the future, we want to further expand the framework of the research system, compare the resilience status of various cities, and examine the driving factors to increase urban resilience and meet the healthy growth of cities in all aspects. In addition, the use of simulation is a method to more intuitively demonstrate the sustainable development of the future city, which is necessary for the later creation of resilience planning that is more accurately positioned.

## 5 Conclusions, recommendations and contributions

### 5.1 Conclusions

This study establishes a comprehensive resilience evaluation of three dimensions of "production space, living space, and ecological space" for the three time points of 2010, 2015, and 2020 using counties located in the Three Gorges Reservoir Area of China as the research object. An all-encompassing resilience evaluation approach that includes the three aspects of "production space", "living space", and "ecological space" was developed. ArcGIS software is used to describe the global spatial analysis, and the score and ranking of the driving factors of the resilience of districts and counties are calculated and analyzed using the geographic probe. In conclusion, the development trend of urban resilience as well as the detection findings of the Geodetector, are used to suggest an optimization plan for urban resilience in the Three Gorges Reservoir Area. From this, the following inferences may be made:

(1) The temporal and spatial evolution characteristics of resilience in the Three Gorges Reservoir Area align with the core concept of urban resilience, which emphasizes the ability of cities to withstand disruptions and adapt to unpredictable hazards. The progressive increase in resilience from 2010 to 2020 indicates that the counties have been enhancing their capacity to absorb and recover from disturbances over time. This finding is consistent with the notion of "evolutionary resilience" [9], which suggests that resilience is not a static state but a dynamic process that evolves through continuous adaptation and learning. The initial low resilience in 2010, when the Three Gorges Project was newly completed, reflects the challenges faced by the counties in adjusting to the new environment and infrastructure. However, as time progressed, the counties' resilience improved, demonstrating their ability to adapt and develop more robust systems in production, living, and ecological spaces.

(2) The spatial distribution of resilience, characterized by a "high on both sides and low in the middle" pattern, can be explained by the uneven distribution of resources and infrastructure. High resilience areas are typically located near major urban centers, where better access to economic opportunities, social services, and environmental management contributes to higher resilience [10]. This pattern is also in line with the findings of other studies that highlight the importance of spatial heterogeneity [17,24]. The spatial clustering in 2010 and 2015, as indicated by the positive Moran's I values, suggests that resilience is influenced by spatial proximity and the diffusion of beneficial practices and resources. However, the shift to a more randomized distribution in 2020 indicates that resilience is becoming more evenly distributed, possibly due to the spread of construction measures across the region.

(3) The identification of key drivers, such as Economic Efficiency, Production Efficiency, and Consumption Level, provides insights into the factors that contribute to urban resilience. These drivers are consistent with the multidimensional nature of resilience, which encompasses economic, social, and environmental aspects [10,16]. Economic and production efficiency are crucial for maintaining the economic vitality of cities, enabling them to withstand economic shocks and support sustainable development [14]. The high explanatory power of these drivers indicates that a robust economy and efficient production systems are fundamental to urban resilience. This finding supports the literature that emphasizes the role of economic resilience as a core component of urban resilience [14,22].The importance of Consumption Level and Transportation Level as drivers highlights the significance of social well-being and infrastructure in enhancing resilience. High consumption levels reflect better living standards and social stability, which are essential for the resilience of urban communities [15]. Efficient transportation systems facilitate the movement of goods and people, enhancing the connectivity and adaptability of cities [17]. These findings are in line with studies that emphasize the role of social and infrastructural resilience in urban systems [15,17].

(4) The interaction analysis of drivers further supports the theoretical understanding of resilience as a complex, interdependent system. The two-factor enhancement and non-linear enhancement interactions indicate that it is not solely

determined by individual factors but by the synergistic effects of multiple factors. This finding aligns with the concept of "system resilience," which emphasizes the interconnectedness and interdependence of different components within a system [8,9]. The strong explanatory power of interactions suggests that a holistic approach, considering the interplay between economic, social, and environmental factors, is necessary for enhancing urban resilience.

## 5.2 Recommendations

Through an investigation of spatiotemporal evolution patterns and a mechanistic analysis of drivers, this study reveals marked variations in resilience capacities among counties in the middle-lower reaches of China's Three Gorges Reservoir Area. It has been recommended that the eight counties that make up the study area should be partitioned. Determined by the division of each county's resilience index, they could be classified into three distinct categories: high-quality resilience zones, medium resilience zones, and low-resilience zones. This would allow for better promotion of the development of the reservoir region and would also improve the overall level of urban resilience. Development recommendations for each categorized zone are systematically presented as follows:

(1) High-quality resilient zones play a leading demonstration role, among which Wanzhou District, Yiling District, and others should fully exploit their urban core competitiveness, lead and stimulate the reservoir area's surrounding counties, and accelerate the creation of the first demonstration experimental area with coordinated and unified development of production-living-ecological spaces. Governments of these counties and districts should optimize the production efficiency, economic efficiency, and production scale, as well as support the rapid, precise, and coordinated growth of manufacturing space. They must improve local people's living consumption level, promote public infrastructure building, and create a comfortable, convenient, and livable living place with smooth and easy transit and reliable communication information. They should also increase geological disaster prevention projects, improve the living environment, increase urban greening, increase pollution control capacity, and promote the creation of ecological space with a beautiful ecological environment, beautiful mountains, and clear water.

(2) Medium resilience zones vary substantially from year to year and are mostly concentrated near major urban locations. They should rely on county development, develop county special industries, rely on the radiation of high-quality resilience zones, introduce the advantageous resources, and unite surrounding counties of the same level to develop collaboratively and improve their own competitiveness. The counties in the Three Gorges Reservoir Area are spread out on both sides of the Yangtze River, and they should take full advantage of their good ecological resources, rely on the vitality of the tourism industry in the main city of Chongqing, vigorously develop the tourism economy, and improve the counties' core competitiveness. For example, Zigui and Xingshan counties should develop their unique "orange" industry, enhance exports, reduce reliance on international commerce, and so on.

(3) Low-resilience zones demonstrate constrained development potential due to disadvantaged geographical positioning and limited urban development spillover effects. As they are generally plagued by simple industrial structure, they should develop new industries based on local characteristics while planning their industrial structures and increasing production efficiency. They should also make full use of the national dividend policy, build good infrastructure, seize the opportunity to revitalize new and innovative industries, strengthen the driving force of science and technology innovation, and force them to improve the efficiency of the industry.

## 5.3 Contributions

The resilience assessment system constructed in this study makes important contributions at the levels of technological innovation and policy guidance, as follows:

At the level of technological innovation, this study abandons the traditional one-dimensional evaluation paradigm and innovatively constructs a multidimensional quantitative evaluation model for the synergistic development of "production-living-ecological" by integrating 30 heterogeneous indicators. The model emphasizes the development characteristics of underdeveloped regions and pays attention to the efficiency and structural balance of the production system, the basic safety level of the living system, and the disaster prevention and mitigation capacity of the ecosystem. For example, factors such as the total foreign trade import and export ratio (C7), the urban registered unemployment rate (C16), and the proportion of area in high geological hazard-prone areas (C29) are included in the evaluation system, which provides a quantifiable decision support tool for risk management in urban development.

In addition, the constructed indicator system is highly adapted to the specific needs of the region. For example, according to the results of the indicator interaction, the internet penetration rate (X6) and the density of monitoring points for early warning group prevention of earthquakes (X9) are two-factor enhancement and belong to the main driving factors. This finding helps to support the establishment of an early warning system integrating "smart agriculture" and "landslide monitoring" in the main navel orange production areas such as Zigui and Fengjie counties, which in turn improves the risk resilience of the supply chain of specialty agricultural products.

At the policy guidance level, this study identifies the key drivers and their interactions, and provides a scientific basis for policy formulation to promote urban resilience. The evaluation results can directly serve the practice of Territorial and Spatial Planning in China's Yangtze River Economic Belt. For example, by analyzing the interaction between fixed asset investment per unit area (X1) and forest coverage rate (X7), the study can guide grassroots governments to formulate policies and systems such as "ecological balance of occupancy and compensation" and "negative list of industries" in industrial transformation. In parallel with industrial expansion, ecological restoration will be carried out to ensure a dynamic balance between economic development and ecological protection, which will support the implementation of the Yangtze River Protection Strategy.

## Supporting information

**S1 Table. Data sources.**
(XLSX)

## Acknowledgments

We would like to thank the reviewers for their valuable comments and suggestions. We thank all the municipal and county-level government departments in the Hubei Section of the Three Gorges Reservoir Area for providing statistical data.

## Author contributions

**Conceptualization:** Honglin Liu, Junjie Zhu.

**Data curation:** Honglin Liu, Junjie Zhu, Ziming He.

**Formal analysis:** Honglin Liu, Junjie Zhu.

**Funding acquisition:** Honglin Liu.

**Methodology:** Honglin Liu, Junjie Zhu.

**Project administration:** Honglin Liu.

**Software:** Honglin Liu, Junjie Zhu.

**Writing – original draft:** Junjie Zhu, Ziming He.

**Writing – review & editing:** Honglin Liu.

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
