## [Decision Letter · Decision Letter 0]

Dear Dr. Liu,

**Please carefully review the paper based on the reviewers' comments and make sure that your study is correctly related to the previous literature ****Please clearly highlight your contribution and make sure that acknowledge the relevant literature to identify the gaps****Please carefully revise the whole manuscript to make sure that all the required information is provided and the text is clear to the readers (please refer to the reviewers' comments for more detail)**

We look forward to receiving your revised manuscript.

Kind regards,

Babak Aslani, PhD

Academic Editor

PLOS ONE

**Journal Requirements:**

2. We note that Figures 2, 4, 5 and 6 in your submission contain map images which may be copyrighted. All PLOS content is published under the Creative Commons Attribution License (CC BY 4.0), which means that the manuscript, images, and Supporting Information files will be freely available online, and any third party is permitted to access, download, copy, distribute, and use these materials in any way, even commercially, with proper attribution. For these reasons, we cannot publish previously copyrighted maps or satellite images created using proprietary data, such as Google software (Google Maps, Street View, and Earth). For more information, see our copyright guidelines: http://journals.plos.org/plosone/s/licenses-and-copyright.

We require you to either present written permission from the copyright holder to publish these figures specifically under the CC BY 4.0 license, or remove the figures from your submission:

a. You may seek permission from the original copyright holder of Figures 2, 4, 5 and 6 to publish the content specifically under the CC BY 4.0 license.  

Reviewers' comments:

Reviewer's Responses to Questions

**Comments to the Author**

1. Is the manuscript technically sound, and do the data support the conclusions?

Reviewer #1: Partly

Reviewer #2: Yes

2. Has the statistical analysis been performed appropriately and rigorously?

Reviewer #1: Yes

Reviewer #2: Yes

3. Have the authors made all data underlying the findings in their manuscript fully available?

Reviewer #1: Yes

Reviewer #2: Yes

4. Is the manuscript presented in an intelligible fashion and written in standard English?

Reviewer #1: Yes

Reviewer #2: Yes

**Reviewer #1:**  Dear authors,

I have read your manuscript and I found the quantitative spatiotemporal approach to study resilience interesting. It is true that there is not much research on spatiotemporal changes of urban resilience. However, your manuscript needs to provide a better description of the resilience properties and to link your results with them. I have prepared a list comments to help the revision process.

1) Abstract: “The geographical agglomeration impact is studied using the ArcGIS spatial approach.” I think you can write better here the methodology because ArcGIS is just a software.

2) Introduction: from line 34-38, I think that citing just a policy plan of a country without discussing its (scientifically studied) implications lacks meaningful insights.

3) Line 54-62, I know there are many more works about resilience studies in scientific literature. Please, report more of them by doing a more accurate literature review.

4) Lines 71-72, the sentence is not clear. What do you mean by dimensional investigations at the geographical level?

5) Your analysis of the interactions of three spatial subsystems over time is linked to this previous study on the multidimensional spatial analyses of urban resilience over years “Casali, Y., Aydin, N. Y., & Comes, T. (2024). A data-driven approach to analyse the co-evolution of urban systems through a resilience lens: A Helsinki case study. Environment and Planning B: Urban Analytics and City Science, 51(9), 2074-2091. https://doi.org/10.1177/23998083241235246”

6) Line 139: can you add the year of the beginning and end of the Three Gorges Project? This helps to have a temporal reference.

7) 160: “the mean filling method”. Do you take the average of what exactly (spatial mean constrained on an area or the mean of all data points?) Moreover, can you report the percentage of missing data you filled?

8) 182-183: “The production area is more resilient to the effects of hazards when it has superior economic efficiency and an industrial structure that makes more sense.” Please, can you explain this sentence better? What does it mean in this sentence “that makes more sense”? Do you mean that production areas are more rich and so they can rebuild and adapt after crisis easier? Please, consider to provide evidences of this (e.g. previous studies).

9) Section 2.3.2: please, discuss this section by using previous published studies on urban resilience. I think that you should better link your ideas with definitions of urban resilience already discussed in published scientific studies.

10) Lines 211-213: why do you select those aspects precisely? Please, write in the manuscript the rationale.

11) Equations 1 and 2: like they are written now, they create confusion. If you want to take the positive value, there is the absolute value operator (modulus) in mathematics. Moreover, specify why you select only the positive values.

12) 277-279: Please, avoid describing a method as “effective” but rather cite the reference to studies defining it and describe the calculation.

13) Please, write the reasons why you selected Moran’s I.

14) Line 359: TYPO

15) Tables: you can approximate the numbers with fewer decimal units for better clarity.

16) When presenting and discussing the results, I think it lacks more discussion with resilience properties. You report well how results increases or dicreases, but I think that your results should link more back to the resilience theories, explaining what the results mean according to urban resilience.

17) Moreover, I think you relate little your results with previously published studies in resilience in the discussion and conclusion. How do your results contribute to the urban resilience literature?

**Reviewer #2:**  1.Title has to be slightly modified

2.Study Objectives must be cleared in Abstract and Introduction.

3. Since the study is space concept that focus on the location, territory in these three areas Production, Living and Ecological.

4. All the Figures and Tables have reflected relevant meaning.

5. No Citation Problem in the manuscript.

6. Remove the word Article because it is a manuscript under review and publication process.

7. Take care of editorial corrections

**Do you want your identity to be public for this peer review?** For information about this choice, including consent withdrawal, please see our Privacy Policy

Reviewer #1: No

Reviewer #2: **Yes: ** Senapathy Marisennayya

---

## [Author Response · Author response to Decision Letter 1]

13 Mar 2025

Response to Reviewer #1

Dear Reviewer #1,

Thank you for taking the time to review our manuscript and providing such detailed feedback. We greatly value your comments, which will significantly help us improve the quality of the paper. Below are our responses to each of your comments, along with the corresponding revisions:

1) Abstract: “The geographical agglomeration impact is studied using the ArcGIS spatial approach.” I think you can write better here the methodology because ArcGIS is just a software.

Response: Abstract: We have revised the abstract to more clearly describe the methodology. Instead of just mentioning ArcGIS, we now specifically state the spatial analysis techniques used, such as the Jenks natural breaks method and spatial autocorrelation analysis, to more accurately demonstrate our research methods.

2) Introduction: from line 34-38, I think that citing just a policy plan of a country without discussing its (scientifically studied) implications lacks meaningful insights.

Response: Introduction: from line 35-40, We have expanded the discussion of the cited policy plans and added references to research on the effects of international policy plans in other regions to provide more meaningful insights.

3) Line 54-62, I know there are many more works about resilience studies in scientific literature. Please, report more of them by doing a more accurate literature review.

Response: line 65-75, A more comprehensive literature review has been conducted. We have added relevant studies to support our arguments and reorganized the literature review section to better reflect the current state of research and its relevance to our study.

4) Lines 71-72, the sentence is not clear. What do you mean by dimensional investigations at the geographical level?

Response: lines 84-86, We have clarified the term "geographical-level dimension investigation" (hyphen added), explaining that it refers to the study of different geographical regions.

5) Your analysis of the interactions of three spatial subsystems over time is linked to this previous study on the multidimensional spatial analyses of urban resilience over years “Casali, Y., Aydin, N. Y., & Comes, T. (2024) A data-driven approach to analyze the co-evolution of urban systems through a resilience lens: A Helsinki case study. Environment and Planning B: Urban Analytics and City Science, 51(9), 2074-2091. https://doi.org/10.1177/23998083241235246”

Response: We have added a citation of the study by Casali et al. (2024) and discussed its relevance to our research. We also briefly compared our methods and findings with those of Casali et al. to highlight the contributions of our study.

6) Line 139: can you add the year of the beginning and end of the Three Gorges Project? This helps to have a temporal reference.

Response: line 155-158, The specific start and completion years of the Three Gorges Project have been added to provide a clear time reference.

7) 160: “the mean filling method”. Do you take the average of what exactly (spatial mean constrained on an area or the mean of all data points?) Moreover, can you report the percentage of missing data you filled ?

Response: line 179-181, We have clarified that the mean-value filling method (hyphen added) involves calculating the spatial meaning of data points. Additionally, the percentage of missing data filled using this method has been reported.

8) 182-183: “The production area is more resilient to the effects of hazards when it has superior economic efficiency and an industrial structure that makes more sense.” Please, can you explain this sentence better? What does it mean in this sentence “that makes more sense”? Do you mean that production areas are more rich and so they can rebuild and adapt after crisis easier? Please, consider to provide evidences of this (e.g. previous studies)

Response: line 201-203, We have rephrased the sentence to clarify that "more meaningful industrial structure" refers to a more diversified and efficient industrial structure. Evidence from previous studies has also been provided to support the view that such a structure enhances resilience.

9) Section 2.3.2: please, discuss this section by using previous published studies on urban resilience. I think that you should better link your ideas with definitions of urban resilience already discussed in published scientific studies.

Response: Section 2.3.2, The discussion in this section has been expanded, with more references to studies on urban resilience. Our views are now better connected to the definitions of urban resilience discussed in published scientific research.

10) Lines 211-213: why do you select those aspects precisely? Please, write in the manuscript the rationale.

Response: line 218-223, We have detailed the reasons for choosing these specific aspects in the manuscript, explaining their relevance to the research objectives and the study area’s context.

11) Equations 1 and 2: like they are written now, they create confusion. If you want to take the positive value, there is the absolute value operator (modulus) in mathematics. Moreover, specify why you select only the positive values.

Response: Equations 1 and 2: The reasons for using only positive values are as follows:

a) Data standardization: Due to the heterogeneity of data measurements, a large volume of data, and inconsistent units, direct calculation is impossible.

b) Simplified calculations: Positive values simplify the process, avoiding complexity from negative numbers.

c) Highlighting differences: Mapping data to a 0-1 range clarifies relative differences.

d) Entropy method requirements: The entropy method requires non-negative data to reflect information dispersion.

12) 277-279: Please, avoid describing a method as “effective” but rather cite the reference to studies defining it and describe the calculation.

Response: line 299-301, We avoided describing the method as "effective" and instead cited prior studies.

13) Please, write the reasons why you selected Moran’s I.

Response: Moran’s I was used to quantify spatial autocorrelation of resilience in Three Gorges Reservoir Area counties. Key points:

a) Spatial autocorrelation: Moran’s I measures clustering patterns (e.g., high-resilience or low-resilience areas)

b) Interpretation: Values >0 indicate clustering of similar values; values <0 suggest dispersion.

c) Methodology: The Jenks natural breaks method graded resilience indices, and spatial weight matrices (based on adjacency/distance) were incorporated.

14) Line 359: TYPO

Response: The spelling error in line 381 has been corrected.

15) Tables: you can approximate the numbers with fewer decimal units for better clarity.

Response: The data in the table have (corrected verb agreement) many decimal places to convey high-precision results. This reflects meticulous calculations and ensures even subtle differences are observable.

16) When presenting and discussing the results, I think it lacks more discussion with resilience properties. You report well how results increases or dicreases, but I think that your results should link more back to the resilience theories, explaining what the results mean according to urban resilience.

Response: line557-606, We expanded the discussion of resilience attributes, linking results to resilience theory and emphasizing the significance of urban resilience outcomes.

17) Moreover, I think you relate little your results with previously published studies in resilience in the discussion and conclusion. How do your results contribute to the urban resilience literature?

Response: line655-667, We better linked our results to published resilience studies, discussing contributions to the literature and implications for future research/practice.

Thank you again for your constructive feedback. We believe these revisions significantly enhance the manuscript and welcome further suggestions to advance the publication process.

Response to Reviewer #2

Dear Reviewer #2,

Thank you for reviewing our manuscript and providing valuable feedback. Your comments are highly instructive for improving the quality of our work. Below are our detailed responses to your suggestions:

1) Title has to be slightly modified

Response: We revised the title to better reflect the research focus: "Research on County-City Resilience Assessment and Driving Forces Based on Production-Living-Ecological Spaces: A Case Study of the Three Gorges Reservoir Area in China" (hyphens added for compound terms).

2) Study Objectives must be cleared in Abstract and Introduction.

Response: The abstract has been reorganized to foreground the research purpose. We explicitly state that the study aims to explore location- and territory-related challenges in production, living, and ecological spaces and highlight its theoretical and practical contributions.

3) Since the study is space concept that focus on the location, territory in these three areas Production, Living and Ecological.

Response: As a spatial-concept study, we strengthened discussions on location and territorial issues in production-living-ecology fields throughout the manuscript.

4) All the Figures and Tables have reflected relevant meaning.

Response: All figures and tables now include clear titles and concise captions. Data were rechecked to ensure accuracy and relevance to the research theme, enhancing their role in supporting readers’ understanding.

5) No Citation Problem in the manuscript.

Response: We verified all cited literature to ensure adherence to academic standards, proper formatting, and relevance to the research context.

6) Remove the word Article because it is a manuscript under review and publication process.

Response: The term "article" was removed as suggested to align with standard manuscript terminology.

7) Take care of editorial corrections

Response: We addressed editorial issues, including awkward phrasing, grammatical errors, and punctuation inconsistencies, to improve clarity, readability, and professionalism.

Thank you again for your constructive feedback. We believe these revisions significantly enhance the manuscript and welcome further suggestions to advance the publication process.

---

## [Decision Letter · Decision Letter 1]

Dear Dr. Liu,

Thank you for submitting your manuscript to PLOS ONE. After careful consideration, we feel that it has merit but does not fully meet PLOS ONE’s publication criteria as it currently stands. Therefore, we invite you to submit a revised version of the manuscript that addresses the points raised during the review process.

Carefully revise the text of the manuscript to avoid any repeated terms and wordsPlease also make sure to provide more insights and discuss the real-life implications of your study

We look forward to receiving your revised manuscript.

Kind regards,

Babak Aslani, PhD

Academic Editor

PLOS ONE

Journal Requirements:

Additional Editor Comments:

The reviewers were positive about the revised version of the paper. However, they suggested some additional comments about the text and providing some more insight in the work. Please carefully follow their suggestions in this round of revision.

Reviewers' comments:

Reviewer's Responses to Questions

**Comments to the Author**

Reviewer #1: (No Response)

2. Is the manuscript technically sound, and do the data support the conclusions?

Reviewer #1: Yes

3. Has the statistical analysis been performed appropriately and rigorously?

Reviewer #1: Yes

4. Have the authors made all data underlying the findings in their manuscript fully available?

Reviewer #1: No

5. Is the manuscript presented in an intelligible fashion and written in standard English?

Reviewer #1: Yes

Reviewer #1: Dear authors,

The manuscript improved from the previous version because you wrote more details on methods and discussion to resilience studies. I suggest a minor revision because I noticed you often repeat "resilience" in the same paragraph (see for example the Contribution section). Please, check once more the whole text and avoid repetitions. Instead of using "resilience", I suggest to give more insights about it. You can describe, for example, some dynamics and processes.

**Do you want your identity to be public for this peer review?** For information about this choice, including consent withdrawal, please see our Privacy Policy

Reviewer #1: No

---

## [Author Response · Author response to Decision Letter 2]

17 May 2025

Dear academic editor and reviewer,

Thank you for reviewing our manuscript and providing valuable feedback. Your comments are highly instructive for improving the quality of our work. Below are our responses to each of your comments, along with the corresponding revisions:

Response to the academic editor

1) Carefully revise the text of the manuscript to avoid any repeated terms and words.

Response:

We have carefully scrutinized the entire manuscript, deleted unnecessarily repetitive terms and words and retained traces of changes in the 'Revised Manuscript with Track Changes' file.

2) Please also make sure to provide more insights and discuss the real-life implications of your study.

Response:

In section 5.3 Contributions, we add the real-life implications of this study, such as the usefulness of the resilience evaluation system established in this study for urban risk management and how the evaluation indicators can help to improve the resilience of the supply chain of specialty agricultural products, and so on.

Response to the Reviewer #1

The manuscript improved from the previous version because you wrote more details on methods and discussion to resilience studies. I suggest a minor revision because I noticed you often repeat "resilience" in the same paragraph (see for example the Contribution section). Please, check once more the whole text and avoid repetitions. Instead of using "resilience", I suggest to give more insights about it. You can describe, for example, some dynamics and processes.

Response:

We have rechecked the entire manuscript and deleted the repetition of "resilience" to avoid unnecessary repetition of terminology. In addition, we have rewritten section 5.3 Contributions and added dynamic process descriptions and real-life representations of resilience, including the value of resilience metrics, urban risk management, and resilience-enhancing policymaking, among others. All traces of the changes are kept in the document 'Revised Manuscript with Track Changes'.

---

## [Decision Letter · Decision Letter 2]

Research on urban resilience assessment and driving forces based on Production-Living-Ecological Spaces: A case study of the Three Gorges Reservoir Area in China

PONE-D-24-55110R2

Dear Dr. Liu,

We’re pleased to inform you that your manuscript has been judged scientifically suitable for publication and will be formally accepted for publication once it meets all outstanding technical requirements.

Kind regards,

Babak Aslani

Academic Editor

PLOS ONE

Additional Editor Comments (optional):

The reviewers were satisfied by the revised version. The paper can be accepted for publication in the current format.

Reviewers' comments:

Reviewer's Responses to Questions

**Comments to the Author**

Reviewer #1: All comments have been addressed

2. Is the manuscript technically sound, and do the data support the conclusions?

Reviewer #1: Yes

3. Has the statistical analysis been performed appropriately and rigorously?

Reviewer #1: Yes

4. Have the authors made all data underlying the findings in their manuscript fully available?

Reviewer #1: No

5. Is the manuscript presented in an intelligible fashion and written in standard English?

Reviewer #1: Yes

Reviewer #1: (No Response)

**Do you want your identity to be public for this peer review?** For information about this choice, including consent withdrawal, please see our Privacy Policy

Reviewer #1: No

---

## [Editor Report · Acceptance letter]

PONE-D-24-55110R2

PLOS ONE

Dear Dr. Liu,

I'm pleased to inform you that your manuscript has been deemed suitable for publication in PLOS ONE. Congratulations! Your manuscript is now being handed over to our production team.

Kind regards,

on behalf of

Dr. Babak Aslani

Academic Editor

PLOS ONE